# The Possibility of Using Genotoxicity, Oxidative Stress and Inflammation Blood Biomarkers to Predict the Occurrence of Late Cutaneous Side Effects after Radiotherapy

**DOI:** 10.3390/antiox9030220

**Published:** 2020-03-07

**Authors:** Samia Chaouni, Delphine Dumont Lecomte, Dinu Stefan, Alexandre Leduc, Victor Barraux, Alexandra Leconte, Jean-Michel Grellard, Jean-Louis Habrand, Marilyne Guillamin, François Sichel, Carine Laurent

**Affiliations:** 1ABTE-EA4651, ToxEMAC, Normandie University, UNICAEN, UNIROUEN, 14000 Caen, France; samia.chaouni@unicaen.fr (S.C.); lecomte.del@gmail.com (D.D.L.); d.stefan@baclesse.unicancer.fr (D.S.); alexandre.leduc@unicaen.fr (A.L.); jl.habrand@baclesse.unicancer.fr (J.-L.H.); francois.sichel@unicaen.fr (F.S.); 2Radiotherapy Department, Hôpital Haut-Lévêque, CHU de Bordeaux, 33600 Pessac, France; 3Radiotherapy Department, Cancer Centre François Baclesse, 14000 Caen, France; 4Medical Physics Department, Cancer Centre François Baclesse, 14000 Caen, France; BARRV@baclesse.unicancer.fr; 5Clinical Research Department, Cancer Centre François Baclesse, 14000 Caen, France; A.LECONTE@baclesse.unicancer.fr (A.L.); JM.GRELLARD@baclesse.unicancer.fr (J.-M.G.); 6IFR ICORE-Flow Cytometry Platform, Normandie University, UNICAEN, 14000 Caen, France; m.guillamin@baclesse.unicancer.fr; 7Cancer Centre François Baclesse, 14000 Caen, France; 8SAPHYN/ARCHADE (Advanced Resource Centre for HADrontherapy in Europe), Cancer Centre François Baclesse, 14000 Caen, France

**Keywords:** radiotherapy, toxicity, skin, prediction, genotoxicity, oxidative stress, inflammation, biomarkers

## Abstract

Despite the progresses performed in the field of radiotherapy, toxicity to the healthy tissues remains a major limiting factor. The aim of this work was to highlight blood biomarkers whose variations could predict the occurrence of late cutaneous side effects. Two groups of nine patients treated for Merkel Cell Carcinoma (MCC) were established according to the grade of late skin toxicity after adjuvant irradiation for MCC: grade 0, 1 or 2 and grade 3 or 4 of RTOG (Radiation Therapy Oncology Group)/EORTC (European Organization for Research and Treatment of Cancer). To try to discriminate these 2 groups, biomarkers of interest were measured on the different blood compartments after ex vivo irradiation. In lymphocytes, cell cycle, apoptosis and genotoxicity were studied. Oxidative stress was evaluated by the determination of the erythrocyte antioxidant capacity (superoxide dismutase, catalase, glutathione peroxidase, reduced and oxidized glutathione) as well as degradation products (protein carbonylation, lipid peroxidation). Inflammation was assessed in the plasma by the measurement of 14 cytokines. The most radiosensitive patients presented a decrease in apoptosis, micronucleus frequency, antioxidant enzyme activities, glutathione and carbonyls; and an increase in TNF-α (Tumor Necrosis Factor α), IL-8 (Interleukin 8) and TGF-β1 (Transforming Growth Factor β1) levels. These findings have to be confirmed on a higher number of patients and before radiotherapy and could allow to predict the occurrence of late skin side effects after radiotherapy.

## 1. Introduction

Radiotherapy (RT) remains, with chemotherapy, the cornerstone of cancer treatment. This treatment concerns about 50% of cancer patients, during their cure whether in a curative, adjuvant or palliative situation [1]. Many technical advances have been made to improve treatments, particularly in terms of ballistics (intensity-modulated RT, stereotaxis, hadrontherapy) in order to deliver more precisely the dose to the target zone, while preserving the surrounding healthy tissues [2,3].

In terms of radiation toxicity, we can distinguish early events occurring during or shortly after treatment from late events which take place in a period of more than 6 months to several years after irradiation [4]. Skin is an organ of choice for the study of these toxicities since it is always crossed by radiations during a RT treatment and is constituted of fast proliferating cells and therefore may be representative of other fast-renewal tissues like intestine. The effects of ionizing radiations on skin are complex because of the great diversity of the cell types involved (keratinocytes, fibroblasts, endothelial cells, etc.). The intensity of skin reaction to ionizing radiations is very variable from one individual to another depending on intrinsic radiosensitivity related to kinetics of cell renewal, proliferation rate, hypoxia etc. [5,6,7]. Radiation exposure induces skin side effects like pigmentation, cutaneous atrophy, telangiectasia, subcutaneous fibrosis or necrosis [4]. This can constitute definitive sequelae of the treatment leading to pain and can also lead potentially to malignant transformation.

Skin side effects after RT are often associated to oxidative stress (OS) and inflammation. OS results from the imbalance between the production of Reactive Oxygen Species (ROS) and the antioxidant defense systems. OS leads to damage to macromolecules such as lipids (peroxidation, etc.), DNA (strand breaks, etc.) and proteins (carbonylation, etc.). These lesions can be responsible of severe cellular damage and cause physiological dysfunction and cell death [8,9]. ROS are produced immediately after irradiation but also by subsequent waves which could, among other things, come from inflammatory phenomena. An increase in the level of ROS can also occur due to hypoxia of the tissues, resulting from an alteration in microvascularization, disturbing the balance between ROS and nitric oxide. OS-related reactions have also been observed in many cases of fibrogenesis [10]. Interestingly, in vitro studies have shown the presence of OS late after irradiation in dermal fibroblasts [11] and endothelial cells of the dermis microvasculature [12]. Among antioxidant defenses, non-enzymatic scavengers like glutathione constitute the first line of defense. Then, enzymatic systems are of major importance like: Superoxide Dismutase (SOD), Catalase (CAT) and Glutathione Peroxydase (GPx). Although the origin of late lesions is still debatable, the role of the OS and the cytokine cascade by the recruitment of the immune system seems to be preponderant in the appearance of late skin side effects after RT. Long-term chronic inflammation after irradiation leads to the appearance of OS waves which can lead to damage to the macromolecules, premature senescence or cell death by apoptosis or necrosis.

Identifying and targeting some biomarkers to predict radiation toxicities and evaluating the biological effects of ROS in order to understand all the mechanisms leading to toxicity is of main interest in the field of radiation oncology. Several biomarkers have been studied and identified on various biological compartments such as urine, skin biopsies or blood. Blood is the most accessible in clinical practice for the implementation of a predictive technique on a larger scale and seems the most promising for the prediction of late skin effects [13,14]. In addition, Núńez et al. showed a correlation between the biomarkers evaluated in the blood and in skin biopsies [15]. Among blood biomarkers, lymphocyte apoptosis was the most used to discriminate patients presenting low or high grade of toxicity in general after RT [5,16,17]. More particularly, concerning late cutaneous toxicity, Azria et al. demonstrated a correlation between a low apoptosis rate of CD8 T lymphocytes after ex vivo irradiation at 8 Gy and the occurrence of late skin toxicity greater than grade 2 [7,18]. The authors validated the RadioInduced CD8 T-Lymphocyte Apoptosis (RILA) test as a tool for the prediction of the risk of breast fibrosis. They extended this test to acute toxicity and others cancers with different thresholds of apoptosis having to be determined and with a limitation due to variations in the protocol due to blood sample collection times leading to changes in RILA values [19]. Other biomarkers were studied in blood as 8-oxodG which was shown to increase in ex vivo irradiated blood serum from patients without skin side effects while no increase was measured in patients with grade 3 or 4 [20]. DNA damage was also assessed by micronucleus [21] and comet assays [22] on peripheral blood lymphocytes but results were contradictory and did not allow to discriminate patients presenting or not skin side effects after irradiation. More generally, if we are not only interested in cutaneous skin side effects, the measurement of double-stranded DNA breaks by γ-H2AX in lymphocytes has shown a correlation between the side effects of RT in general and a reduced DNA repair capacity [23,24]. Sprung et al. reviewed a list of cytokine profiles developed in various experimental studies that can be used to predict radiation toxicity [14]. Inflammation was also assessed by measuring C-Reactive Protein (CRP) in blood showing a higher level in patients with early skin side effects greater than grade 2 [25] and should be evaluated later after treatment. Most of the other studies on predicting the occurrence of side effects after RT relate to pulmonary side effects after RT treatment of lung cancer [26,27]. If the specific lung markers are eliminated, the remaining markers tested for their predictive value are cytokines levels, activities of antioxidant enzymes, polymorphism of ATM (Ataxia Telangiectasia Mutated) and TGF-β1 and proteomic analysis of plasma. Indeed, levels of TGF-β [28,29], IL-1α and IL-6 [30], IL-8 [31] and ICAM-1 [32] allowed to discriminate patients developing radiation pneumonitis. Concerning antioxidant enzymes, patients developing radiation pneumonitis showed higher SOD activity and lower GPx activity in erythrocytes compared to patients without side effects [33]. The polymorphism of ATM measured in the leukocytes of patients made it possible to associate it with an increased risk of radiation pneumonia [34] whereas the polymorphism of TGF-β1 could not be associated with an increased risk of pulmonary side effects [35,36]. Finally, proteomic analysis of plasma has shown that certain proteins are overexpressed in patients with high pulmonary toxicity [37,38] and a new method of analysis has revealed that α-2-macroglobulin can significantly dissociate patients with or without pulmonary side effects [39]. Genomic and proteomic approaches are being developed, knowing that the variability of cell types, locations, patients, RT protocols and study protocols remains a limiting factor.

The most common locations for studies of skin side effects are breast tumors. Indeed, Rodriguez-Gil et al. [25] showed that, 6 weeks after conventional RT treatment for breast cancer, 50% of patients had early side effects greater than grade 2. However, the treatment of mammary tumors is often combined, RT being generally associated with chemotherapy, and the dosimetry is not very fine. Our goal being to observe the interindividual variations of the responses to the RT, we had to avoid the parameters which could influence these responses. The ideal location for this study is therefore Merkel Cell Carcinomas (MCC), since the treatment with RT is not combined to another, the surface of irradiated skin is extensive and the dosimetry is precise. MCC is a rare aggressive neuroendocrine skin cancer that occurs mainly in previously photodamaged skin in older people [40]. In this way, knowing that aging, as well as photoaging, influences the response of skin to RT, age of the patients must be taken into account [41], as well as skin phototype which was shown to influence the response to RT [42]. MCC is located principally in the head/neck (48–53%) and extremities (34–35%) [43,44]. Current treatments are mainly surgery, RT and more recently immunotherapy with the use of Avelumab which showed high efficiency against metastatic MCC [45]. 

For this study, patients treated by adjuvant RT for MCC were separated into groups according to the grade of late cutaneous toxicity developed. This parameter was assessed depending on the Radiation Therapy Oncology Group (RTOG) and the European Organization for Research and Treatment of Cancer (EORTC) [46]. Two groups were established: (i) patients with no or little toxicity (grade 0, 1 or 2 of the RTOG) and (ii) patients with marked toxicity (grade 3 or 4 of the RTOG). The aim of our study was to highlight one or more blood biomarkers of apoptosis, genotoxicity, OS and inflammation, the variations of which after ex vivo irradiation could be used to predict the occurrence of late skin side effects after RT.

## 2. Materials and Methods 

### 2.1. Reagents

RPMI and FBS were obtained from Thermo Fisher Scientific (Waltham, MA, USA), DNA-Prep Reagent Kit from Beckman Coulter (Brea, CA, USA), NADPH from Roche (Mannheim, Germany), Legendplex™ Human Inflammation Panel from BioLegend (San Diego, CA, USA) and MILLIPLEX^®^ MAP Kit and Superoxide Dismutase Assay kit II from Merck (Kenilworth, NJ, USA). All other chemicals were purchased from Sigma-Aldrich (Saint-Louis, MO, USA).

### 2.2. Patients

Among the recruited patients treated by RT for MCC at the Cancer Center François Baclesse, two groups of 9 patients were constituted according to their skin toxicity grade. One of these groups called “Tox ≥ 3” presented extensive skin lesions (grade 3 or 4 of the RTOG/EORTC). The other group called “Tox ≤ 2” presented no or slight skin side effects (grade 0, 1 or 2 of the RTOG/EORTC). The procedures of this study were reviewed and approved by the committee of protection of person (2016-A02021-50; approval date: 01/31/2017). The collection of clinical and dosimetric data was carried out during a follow-up consultation. Late skin toxicity was assessed after signing an informed consent. Blood samples were collected on 6 heparinized tubes. Possible confounding factors were collected: diabetic status, high blood pressure, vitamins and antioxidants taking, treatments in progress at the time of inclusion and at the time of treatment, body mass index or BMI, phototype according to the classification of Fitzpatrick and any other intercurrent pathologies. 

### 2.3. Irradiation

Irradiation of blood samples was performed by 6 MV photon beams from an ARTISTE linear accelerator (Siemens) at room temperature in the RT department of Cancer Center François Baclesse. The dose rate used was the standard rate in conventional treatment: 2 Gy/min. The delivered doses were 2 and 10 Gy and corresponded respectively to 290 and 1450 MU (monitor units). The unirradiated control blood samples were transported under the same conditions to the RT department but were not irradiated. At the end of the irradiation, samples were placed for 1 h in a cell culture incubator at 37 °C and 5% CO_2_ with a controlled humidity level.

### 2.4. Separation of Blood Components

A total volume of 8 mL per irradiation dose (0, 2 and 10 Gy) was collected. For each condition, 3 mL were used for lymphocyte cycle analysis. The remaining 5 mL per dose per patient was placed in tubes and centrifuged to separate the different blood components. For cycle and apoptosis, samples were diluted in 3 mL of PBS and then divided into 3 cytometry tubes each containing 1.5 mL of Ficoll. The triplicate samples were centrifuged at 400× *g* for 35 min (room temperature, no brake) to separate the different blood components and recover a clean lymphocyte ring. For other measurements, blood components were separated by centrifugation only. After 10 min of centrifugation at 1100× *g* at room temperature, 3 compartments were obtained and treated as followed: (i) the plasma was aliquoted and frozen at –80 °C, (ii) the lymphocyte ring was used for micronucleus study and (iii) the erythrocyte pellet was aliquoted in the same way as the plasma and frozen at –80 °C.

### 2.5. Lymphocyte Cycle and Apoptosis 

Lymphocyte rings were cultured in RPMI 1640 medium supplemented with 10% Fetal Bovine Serum (FBS) and 1% penicillin/streptomycin. The culture was initiated by adding 60 µM of phytohemagglutinin (PHA) on a 6-well plate and placed in an incubator at 37 °C and of 5% CO_2_ in a humid atmosphere. After 48 h of incubation, the content of each culture well was transferred into a tube and then centrifuged at 150× *g* for 5 min at room temperature. After washing with PBS and centrifugation, cells were fixed in 70% ethanol and stored at –20 °C. Before flow cytometry analysis, alcohol was removed and cells were washed with PBS and incubated at 37 °C during 30 min. After 5 min centrifugation at 2700× *g*, the DNA-Prep Reagent Kit was used according to the manufacturer’s recommendations. In brief, cells were resuspended in the presence of Lysing Permeabilizing Reagent (LPR) then marked with STAIN solution containing RNase and Propidium Iodide (PI). Cells were incubated at room temperature during 20 min in the dark and transferred in specific cytometry tubes for analysis. Sample analysis was performed using a Gallios flow cytometer (Beckman Coulter, Brea, CA, USA) within one week after sample collection. The fluorescence of IP was collected in the FL3 channel with 620 nm bandpass filter. The singulets were selected using an area versus peak DNA content histogram and then analysed in a single-parameter histogram FL3 lin. The cycle data were analyzed according to the distribution of the lymphocytes in the different phases: sub-G1 corresponding to the apoptotic cells, G1 phase, G2-M phase and S phase. Data were then acquired with Gallios software and analysed with Kaluza software (Beckman Coulter). 

### 2.6. Micronucleus Frequency in Lymphocytes

The lymphocyte ring of each sample was put in 6-well plates containing RPMI 1640 culture medium (10% Fetal Bovine Serum or FCS, 1% penicillin/streptomycin) in addition to 60 μM of phytohemagglutinin (PHA) to stimulate cell division of T cells. Plates were incubated at 37 °C and 5% CO_2_ in a humidified atmosphere. After 44 h of incubation, cytochalasin B was added to a final concentration of 5 μg/mL in each culture well to block cytokinesis. After an additional 28 h of incubation, the content of each well was transferred to a FACS tube, centrifuged at 180× *g* for 10 min at room temperature. Lymphocytes pellets were then subjected to a hypotonic shock by adding 75 mM KCl dropwise over 10 min. The content of each pellet was finally spread on 3 slides on humidified paper towels before drying and storage at –20 °C. For the analysis, labeling was carried out by adding 40 μL of mounting medium containing DAPI and a coverslip on each slide. Analysis was performed under 10× magnification using an automated scoring system Metafer (MetaSystems, Altlussheim, Germany) coupled with a fluorescence microscope (Zeiss) in order to quantify the micronuclei frequency and distribution by binucleated lymphocytes.

### 2.7. Lysis of Erythrocytes 

Erythrocytes were placed in a lysis buffer (10 mM Tris/HCl, 0.1% Triton, 200 mM sucrose, pH 7.5) and lysed by thermal shock: 1 min in liquid nitrogen and 1 min at 37 °C. This step was repeated 3 times, then samples were centrifuged at 4 °C at 2000× *g* for 30 min. The supernatants obtained were aliquoted in Eppendorf tubes and stored at –80 °C.

### 2.8. Protein Quantification in Erythrocyte Lysates 

Protein assay was performed by using a protein quantification Kit-Rapid according to the manufacturer’s recommendations. In brief, 6 µL of diluted samples were mixed with 300 µL of Coomassie Brilliant Blue, then the absorbance was measured at 595 nm using a microplate reader (Fluostar Omega, BMG Labtech, Ortenberg, Germany). Protein concentration was determined using the standard curve equation.

### 2.9. SOD activity in Erythrocyte Lysates 

Total SOD activity was measured using Superoxide Dismutase Assay kit II according to the manufacturer’s recommendations. The SOD assay relies on the detection of superoxide radicals which are generated by xanthine oxidase and hypoxanthine via a tetrazolium salt. The absorbance was measured at 450 nm by using a Fluostar Omega microplate reader (BMG Labtech, Ortenberg, Germany). The standard curve established with different concentrations of standards was used to calculate the SOD activity. Results are expressed in units per mg of proteins. 

### 2.10. CAT Activity in Erythrocyte Lysates

CAT activity was assayed using the Clairborne and Aebi spectrophotometric method [47]. In brief, erythrocyte lysates were diluted in a 50 mM potassium phosphate buffer (99:1, *v:v*). Twenty-five microliters of diluted samples were dispensed in UV-star plates (Greiner Bio-One, Kremsmünster, Austria). The reaction was initiated by the addition of 225 µL of hydrogen peroxide (30 mM). The decrease in absorbance at 240 nm was monitored by a Flexstation 3 microplate reader (Molecular Devices, San Jose, CA, USA) for 1 min. CAT activity was calculated using slope values from standard bovine purified liver CAT. Results are expressed as nmol of consumed hydrogen peroxide per min per mg of proteins.

### 2.11. GPx Activity in Erythrocyte Lysates 

GPx activity was assayed using Sinet method with slight modifications [48]. Briefly, erythrocyte lysates were diluted (1/625) in a buffer containing 125 mM potassium phosphate buffer (Na_2_HPO_4_/NaH_2_PO_4_, pH 7), 12.5 mM EDTA, 50 mM KCN, 5 mM reduced glutathione, 5 mM NADPH and 0.25 UI of Glutathione Reductase. Diluted samples were distributed in 96-well plates and incubated at 30 °C during 15 min. Reaction was initiated by the addition of 250 µM of Tert Butyl Hydroperoxydase. The decrease in absorbance at 340 nm was monitored with a Fluostar Omega microplate reader (BMG Labtech) for 2.5 min. GPx activity was determined by the calculation below and results were expressed as nmol of oxidized GSH (GSSG) per min per mg of proteins.
$$\mathrm{GPx}\;\mathrm{Activity}=2\times \frac{\frac{\left|\mathrm{slope}\right|\;\left(\mathrm{AU}.\min^{-1}\right)}{\varepsilon_{\mathrm{NADPH}}\left({\mathrm{cm}}^{-1}\right)\times l\;\left(\mathrm{cm}\right)}\times \frac{\mathrm{Vf}\;\left(\mathrm{mL}\right)}{\;\mathrm{Vs}\;\left(\mathrm{mL}\right)}\times \mathrm{sample}\;\mathrm{dilution}}{\mathrm{protein}\;\mathrm{concentration}\;\left(\mathrm{mg}/\mathrm{mL}\right)}$$

AU: Absorbance Unit, ε: molar extinction coefficient for NADPH (Nicotinamide Adenine Dinucleotide Phosphate) at 340 nm (0.00622 µm^–1^.cm^–1^), l: optical path length, Vf: final volume per well, Vs: volume of diluted sample 

### 2.12. Quantification of Reduced and Oxidized Glutathione in Erythrocyte Lysates 

A quantification kit was used to measure oxidized and reduced glutathione according to the manufacturer’s recommendations. The amounts of total glutathione (reduced and oxidized) were determined by an enzymatic method. Briefly, GSSG (glutathione disulfide, oxidized form) was first reduced to GSH (reduced form of glutathione) by the addition of NADPH and glutathione reductase in controlled amounts. 5,5’-dithiobis-2-nitrobenzoic acid reacts with GSH to form a product detectable by spectrophotometry at 412 nm. To assay only oxidized glutathione, a masking reagent was added to the samples to trap the initial GSH. Levels of GSH and GSSG in samples were measured by a microplate reader (Model 680, Bio-Rad, Hercules, CA, USA) and calculated from GSH and GSSG standard curve respectively. Results are expressed in µmol of GSH or GSSG per mg of proteins.

### 2.13. Protein Carbonylation in Erythrocyte Lysates 

Protein carbonylation measurement was performed using a protein carbonyl content assay kit according to the manufacturer’s recommendations. In brief, erythrocyte lysates were first derived in dinitrophenyl (DNP) hydrazone adducts in the presence of 2,4-dinitrophenylhydrazine (DNPH). The addition of trichloroacetic acid allowed the precipitation of proteins. An acetone washing step was carried out to remove excess DNPH and to retain only the proteins. After centrifugation, pellets were suspended in a 6 M guanidine solution. The absorbance was measured at 375 nm by a Fluostar Omega microplate reader (BMG Labtech, Ortenberg, Germany). The amount of protein carbonyls was calculated using the formula below. Results are expressed as nmol of carbonyls per mg of proteins.
Protein carbonyls (nmol carbonyls/mg protein) = (C/P) × 1000 × D

C: amount of carbonyls in sample wells (nmol/well), P: amount of proteins from standard wells, D: dilution factor of samples, 1000: conversion factor (µg to mg) 

### 2.14. Lipid Peroxidation in Erythrocyte Lysates

Lipid peroxidation was measured using PeroxiDetect Kit according to the manufacturer’s recommendations. Typically, peroxides react with Fe^2+^ ions to produce Fe^3+^ ions in the same proportion to the amount of hydroperoxides initially present in samples. The Fe^3+^ ions then react with the xylenol orange (3,3′-bis[N,N-bis(carboxymethyl)aminomethyl]-o-cresolsulfonephtalein, sodium salt) and form a colored compound which can be detected by spectrophotometry at 570 nm. The amount of lipid hydroperoxide was measured by a microplate reader (Model 680, Bio-Rad) and calculated from the standard curve of tert-Butyl Hydroperoxide (t-BuOOH). Results are expressed as nmol of peroxides per mg of proteins.

### 2.15. Quantification of Inflammatory Cytokines in Plasma by Flow Cytometry 

#### 2.15.1. LegendplexTM Human Inflammation Panel

A panel of 13 cytokines (IL-1ß, IFN-α2, IFN-y, TNF-α, MCP-1, IL-6, IL-8, IL-10, IL-12p70, IL-17A, IL-18, IL-23, IL-33) was quantified in plasma using Legend plexTM Human Inflammation Panel according to the manufacturer’s recommendations. In brief, fluorescent encoded beads coated with specific antibodies on their surface capture the analytes. After an incubation and a washing step, the biotinylated detection antibody is added and binds to its specific analyte. Finally, streptavidin-phycoerythrin (SA-PE) is used as a reporter. Flow cytometry analysis was performed using a CytoFlex flow cytometer (Beckman Coulter, Brea, CA, USA). The configuration used was 585 nm with a blue laser (FL-1A) and 690 nm with a yellow laser (FL-9A). Finally, data were analyzed using LEGENDplex™ version 8 data analysis software (BioLegend, San Diego, CA, USA).

#### 2.15.2. Milliplex^®^ MAP Kit: TGF-ß1 Single Plex Magnetic Bead Kit 

TGF-ß1 cytokine was quantified using MILLIPLEX^®^ MAP Kit according to the manufacturer’s recommendations. Briefly, fluorescent magnetic beads coated with TGF-ß1 antibodies capture the analyte, then a biotinylated detection antibody binds to this complex and SA-PE (Streptavidin Phycoerythrin) is used as a reporter. The MAGPIX reader (Luminex corp, Avatia, TX, USA) was used to capture images. Finally, data were analyzed using xPONENT^®^ software (Luminex corp, Avatia, TX, USA).

### 2.16. Statistical Analysis 

Data are depicted as mean values ± standard error of the mean (SEM). * for *p* < 0.05, ** for *p* < 0.001 and *** for *p* < 0.0001 for irradiated samples compared to non-irradiated ones and # for *p* < 0.05 and ## for *p* < 0.005 for Tox ≥ 3 compared to Tox ≤ 2 group (two-way ANOVA and Mann–Whitney test). Each experiment was performed in triplicates.

## 3. Results

### 3.1. Patient Profiles 

Eighteen patients were included in this study (Table 1). Each group contained nine patients according to their RTOG late skin toxicity classification. The average age of the patients included was around 70 years old as expected for the MCC pathology characterized by elderly patients. There were no significant differences in terms of received irradiation dose, BED (Biological Effective Dose with α/β = 3), cardiovascular risk factor number (diabetic status, hypertension, and obesity with a Body Mass Index ≥ 30, in addition to age), tobacco consumption or grade of acute toxicity after RT (data not shown). Late skin toxicity was assessed by a junior and a senior radiation therapist based on clinical observations and according to RTOG/EORTC classification (Figure 1).

### 3.2. Cell Cycle and Apoptosis

Percentage of lymphocytes in each phase of the cell cycle was evaluated by flow cytometry. In both groups of patients, the rate of sub-G1 phase (corresponding to apoptosis) increased with irradiation (Figure 2a). This increase was only significant for the Tox ≤ 2 group with a 1.8-fold change at 10 Gy. Moreover, a significant difference exists between the two groups of patients with a 1.25-fold decrease in the Tox ≥ 3 group compared to the Tox ≤ 2 group. The proportion of cells in S phase was significantly reduced after 10 Gy irradiation in the same manner for both groups with a 1.41-fold and a 1.71-fold decrease in the Tox ≤ 2 group and in the Tox ≥ 3 group, respectively (Figure 2b). There is no significant difference concerning the G2-M phase for both groups even if a tendency towards a decrease in the percentage of lymphocytes in G2-M phase has been observed after irradiation. When comparing the two groups of patients, a non-significant increase was measured in the Tox ≥ 3 group compared to the Tox ≤ 2 group even in unirradiated blood samples (Figure 2c). Regarding the G0-G1 phase, no difference was observed after irradiation or between both groups of patients (data not shown). Figure 2d illustrates the modification of the cell cycle profile of a patient of the Tox ≥ 3 group after irradiation at 10 Gy.

### 3.3. Micronucleus Assay 

Genotoxicity was assessed by the micronucleus assay. Micronuclei contain a chromosome or fragment(s) of chromatid(s) which have not been incorporated into one of the daughter nuclei after cell division. Micronucleus assay has been used to evaluate poorly repaired or unrepaired DNA breaks or nondisjunction of chromosomes. A significant increase in micronucleus frequency was observed with irradiation as expected (Figure 3a and Table 2) with a 81-fold increase in MN frequency in irradiated compared to non-irradiated lymphocytes in the Tox ≤ 2 group compared to a 17-fold increase in the Tox ≥ 3 group. This difference between the two groups of patients was significant with a 3.31-fold decrease in the Tox ≥ 3 compared to the Tox ≤ 2 group. These results were confirmed by examining the distribution of micronuclei with an increase in each class of the distribution of micronuclei in the Tox ≤ 2 compared to the Tox ≥ 3 group (Figure 3b and Table 3).

### 3.4. Antioxidant Enzyme Activities in Erythrocyte Lysates

Measuring the activity of the three potent enzymes (SOD, GPx and CAT) gives an idea of the antioxidant capacity of erythrocytes since these enzymes are the second line of the enzymatic antioxidant defense chain after non-enzymatic antioxidant components such as vitamins E/C and some small molecules like ubiquinon and glutathione. Differences were observed between the two groups of patients with decreasing trends for the three enzymes of the Tox ≥ 3 compared to the Tox ≤ 2 group (Table 4). With regard to SOD, this trend was significant, reaching a 1.47-fold decrease at 2 Gy.

### 3.5. Reduced and Oxidized Glutathione in Erythrocytes Lysates

The measurement of reduced and oxidized glutathione levels allows evaluating the cellular oxidative state. Reduced and oxidized glutathione levels were analyzed. Levels of reduced and oxidized glutathione have remained statistically unchanged in the global population with or without irradiation (Figure 4). However, trends towards a decrease in GSH (Figure 4a) and GSSH (Figure 4b) were observed in the Tox ≥ 3 compared to the Tox ≤ 2 group reaching a 1.24-fold decrease concerning oxidized glutathione. In addition, the GSH/GSSG ratio was 1.14-fold increased in the Tox ≥ 3 compared to the Tox ≤ 2 group at 10 Gy (Figure 4c).

### 3.6. Protein Carbonylation and Lipid Peroxidation in Erythrocytes Lysates

The quantification of protein carbonylation and lipid peroxidation in erythrocyte homogenates was analyzed to assess lipid and protein oxidation in blood samples from both groups of patients. No significant difference was observed with the irradiation, nor according to the group of toxicity (Table 5). A tendency to a decrease in protein carbonylation was observed in irradiated blood of the most radiosensitive patients compared to the less radiosensitive ones with a 1.18-fold decrease in the Tox ≥ 3 group compared to a 1.02-fold increase in the Tox ≤ 2 group (10 Gy vs 0 Gy).

### 3.7. Inflammatory Cytokines in Plasma

The plasma levels of 14 cytokines involved in the mechanisms of inflammation and the regulation of the immune system response were tested in both groups of patients at 0 and 10 Gy. The levels of IL-1/6/10/12p70/17A/23/33 and IFN-α2/y were generally below the detection threshold (data not shown). The MCP-1 and IL-18 levels were above the detection threshold but without any significance or any trend after irradiation or between both groups of patients (data not shown). Regarding TNF-α, it should be noted that a non-significant increase was observed after irradiation with a tendency to decrease in the Tox ≥ 3 compared to the Tox ≤ 2 group reaching a 2.54-fold change (Figure 5a). IL-8 plasmatic concentration was strongly, but not significantly, increased in the Tox ≥ 3 compared to the Tox ≤ 2 group, even in the non-irradiated blood samples with a 2.49-fold change at 0 Gy and a 1.58-fold change at 10 Gy (Figure 5b). Interestingly, there is no strong effect of irradiation on IL-8 level. In the same way, TGF-β1 plasmatic concentration was significantly increased in the Tox ≥ 3 compared to the Tox ≤ 2 group with a 1.32-fold change at 0 Gy and a 1.61-fold change at 10 Gy (Figure 5c). Moreover, irradiation tends to decrease TGF-β1 level in both groups with a 1.40-fold decrease after irradiation for the Tox ≤ 2 group and a 1.15-fold decrease after irradiation for the Tox ≥ 3 group.

## 4. Discussion

The aim of this study was to assess biomarkers of genotoxicity, OS and inflammation in blood of MCC patients in order to discriminate patients who develop severe or mild late cutaneous side effects after RT. For this purpose, blood samples from two groups of nine patients, constituted according to their toxicity grade, were collected and irradiated ex vivo.

The patient’s late skin toxicity grade was evaluated as described by RTOG/EORTC knowing that (i) slight atrophy, change in pigmentation, hair loss and moderate telangiectasia were classified as grade 1 or 2 and (ii) marked atrophy, ulceration and strong telangiectasia were classified as grade 3 or 4 (Figure 1). Skin grafts were performed on the nine patients in the Tox ≤ 2 group and on 5/9 patients in the Tox ≥ 3 group. The 18 patients recruited for this study completed a survey before blood collection in which different parameters were taken into account as the location of the tumor, the total dose received during treatment, the biological effective dose, the development of acute and/or late toxicities, Fitzpatrick phototype as well as age, cardiovascular diseases (diabetes, hypertension and obesity) and tobacco consumption. Among these patients there were no significant difference between the 2 groups (Table 1).

As expected, lymphocyte apoptosis was increased after irradiation (Figure 2). As shown by Azria et al. [18], this increase was higher in patients presenting lower late skin toxicity. These results suggest that lymphocytes of patients with higher late skin toxicity cannot undergo apoptosis as are patients with less radiosensitivity. This difference in apoptosis in both groups of patients could be linked to the TRAIL (TNF-Related Apoptosis Inducing Ligand) pathway. Indeed, a higher expression of TRAIL has been observed in radiosensitive T4 effector memory lymphocytes compared to radioresistant ones [49]. Further studies are needed to investigate the role of TRAIL-R knowing that TRAIL-R1 or -R2 are in favor of a pro-apoptotic pathway while TRAIL-R4 is related to an anti-apoptotic pathway via NF-κB [50,51]. The lower rate of apoptosis in patients with higher toxicity may be correlated with a non-significant increase in G2-M arrest in these patients. Interestingly, this trend to an increase in the G2-M phase was observed even in unirradiated samples. However, probably due to the small number of patients, these changes were not statistically significant. In addition, the results concerning lymphocyte micronucleus frequency showed a significant lower increase after irradiation in the most radiosensitive patients (Figure 3a and Table 2). This could be related to a misrepair of DNA damage leading to fewer micronuclei but bringing potentially stochastic effects. Lymphocytes of the least radiosensitive patients, presenting more micronuclei resulting from DNA damage and repair, could continue to progress in the cell cycle with less G2-M arrest for DNA repair, possibly inhibited by the p21 pathway, to finally undergo apoptosis as shown by sub-G1 cell percentage analysis. In contrast, lymphocytes from the most radiosensitive patients have fewer micronuclei perhaps due to the elimination of the most damaged cells by mitotic death. These results were consistent with the study of Skiöld et al. on the oxidized base 8-oxodG which was increased in ex vivo irradiated blood only in patients presenting no skin side effects [20].

Blood antioxidant capacity was assessed by the measurement of SOD, CAT, GPx, GSH and GSSG in erythrocytes as red blood cells are considered as the main actors of blood ROS elimination [52]. After irradiation, the antioxidant levels in erythrocytes decreased in the most radiosensitive patients. This reduction concerned GPx, CAT, glutathione but was only significant for SOD which plays a central role in the detoxification of ROS (Table 4 and Figure 4). These trends were observed for almost all antioxidants even in unirradiated samples. Results obtained at the basal level (i.e., unirradiated blood) are difficult to compare with the data in the literature because of the different methods used and the variety of expression of the results (per volume, per mg of proteins or hemoglobin etc.). Although OS has been extensively studied in various pathologies such as diabetes, Alzheimer’s disease, and in aging, there are no validated data of refence levels in blood. The decrease in SOD level in the most radiosensitive patients could not be correlated with the results observed concerning early pukmonary side effects after irradiation in C3H/HeN mice: an increase in erythrocyte SOD activity was linked to an increase in radiation pneumonitis [33]. However, Park et al. [33] also demonstrated a decrease in the activity of glutathione peroxidase in human erythrocytes which is in agreement with our results. The slight, non-significant decrease in GPx activity in the most radiosensitive patients was correlated with a slight, non-significant increase in the GSH/GSSG ratio. It might be interesting to assess the activity of glutathione reductase to explain this GSH/GSSG ratio. The level of lipid peroxidation and protein carbonylation was statistically unchanged in the two groups of patients with only a tendency to decrease carbonyls after irradiation in the most radiosensitive patients (Table 5). Altogether, these results suggest that the most radiosensitive patients seemed to present a decrease in defense mechanisms resulting in a decrease in ROS detoxification.

The previous results are consistent with what has been observed concerning inflammation (Figure 5). A significant increase in TGF-β1 was measured in plasma samples from the most radiosensitive patients. This increase could explain an inhibition of a pro-apoptotic pathway via ATM and NF-κB, knowing that TNF-α was not significantly increased in these patients, so that TRAIL-R4 could be of major interest to study. Moreover, IL-8, which was not significantly increased in the most radiosensitive patients, is known to induce angiogenesis and tissue remodeling [53]. An increase in inflammation, measured by the blood level of CRP (C-reactive protein) in the most sensitive patients to early skin side effects of RT, was also observed by Rodriguez-Gil et al. [25]. It is also known that localized irradiation can lead to a late increase in inflammatory status. As shown by Liao et al. on C57Bl6 mice 4 weeks after irradiation, IL1β and IL23 were strongly upregulated especially IL-17, which is responsible for upregulating γδ T cells involved in mediating innate immune response, playing an important role on skin inflammation and thus on the appearance of radiation dermatitis [54]. Since inflammation is also a source of ROS, the decrease in antioxidant capacity in the most radiosensitive patients could not help to detoxify the overproduction of ROS in these patients. Interestingly, the basal rate of plasmatic cytokines was higher in radiosensitive patients suggesting that these patients present a higher inflammatory status than the least radiosensitive ones. Aging, which make patients more vulnerable to RT, could also be critical point to take into consideration as it could influence inflammatory response [55]. This difference in basal rates between both groups of patients can be correlated with the study of Anscher et al. [28] in which they demonstrate that TGF-β1 rate was higher in the group of patients who developed pneumonitis after RT. It has to be noted that IL-6 level was below the detection threshold even in irradiated blood which is surprising compared to previous published results [56]. Moreover, as reviewed by Mavragani et al. [57], DNA damage/repair and chronic inflammation powered by the induction of DNA damage response are the most important factors to take into account for the prediction of radiation effects.

## 5. Conclusions

In conclusion, the results that we obtained on 18 patients showed a high interindividual variability in radiosensitivity between patients as expected. The use of skin biopsies would be more relevant, especially concerning DNA damage [58,59], inflammation [60], survival [61] and gene expression profile [62,63] but difficult to obtain from every patients before their RT and the methods would be more arduous to implement on a large scale. Moreover, skin aging, especially in the case of MCC patients, should be evaluated before RT. In this way, targeting dermal fibroblasts, especially myofibroblast by using for example alpha-SMA (Smooth Muscle Actin) as a biomarker could give an indication on the microenvironment of the tissue as well as its role on biological radiation effects [64,65]. Nevertheless, this study allowed us to highlight differences between patients with high or low grades of late skin side effects after RT. In patients presenting low grades of late skin radiation toxicity, cell death by apoptosis or mitotic death associated to high micronucleus frequency may help to eliminate the damaged cells. In addition, the most radiosensitive patients present a decrease in antioxidant capacity and an increase in inflammatory cytokines. Therefore, the use of a group of biomarkers seemed to be more relevant to ensure a better prediction of side effects of RT. To confirm these findings and to improve the statistical power in order to establish prediction models, it is of major importance to enlarge the number of patients. The ultimate goal would be to extend the study to other tumor sites than MCC. This will undoubtedly be complex since, for example, age differences between patients will result in different baseline levels of biomarkers of OS. Furthermore, according to the results obtained, it would be interesting to investigate DNA repair systems and signaling pathways leading to apoptosis, as well as transcriptional and post-translational modifications which could explain differences observed in OS and inflammation biomarkers. Concerning DNA damage, Mavragani et al. [57] pointed out the major role of clustered DNA damage and the use of Monte-Carlo prediction model but also the interest of using bioinformatics and omics approaches. Currently, genomic and proteomic approaches are in full development knowing that the variability of cell types, locations, patients, RT protocols remains a limiting factor. Finally, the discrimination of patients who will or will not experience side effects of RT could allow (i) to prevent these effects by adapting their treatment before RT, (ii) to treat side effects by using treatments during and after RT [66,67] or (iii) to use these knowledge to improve treatments [68].

## Figures and Tables

**Figure 1 antioxidants-09-00220-f001:**
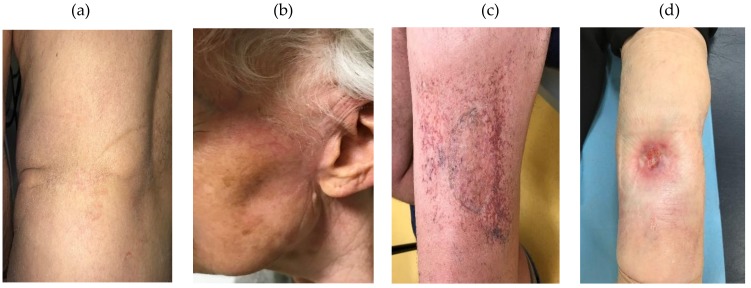
Pictures of some patient irradiated skin regions: (**a**) patient with grade 0 toxicity (no fibrosis, no telangiectasia); (**b**) patient with grade 1 toxicity (no fibrosis, imperceptible telangiectasia); (**c**,**d**) patients with grade 3 toxicity (pronounced fibrosis, marked telangiectasia).

**Figure 2 antioxidants-09-00220-f002:**
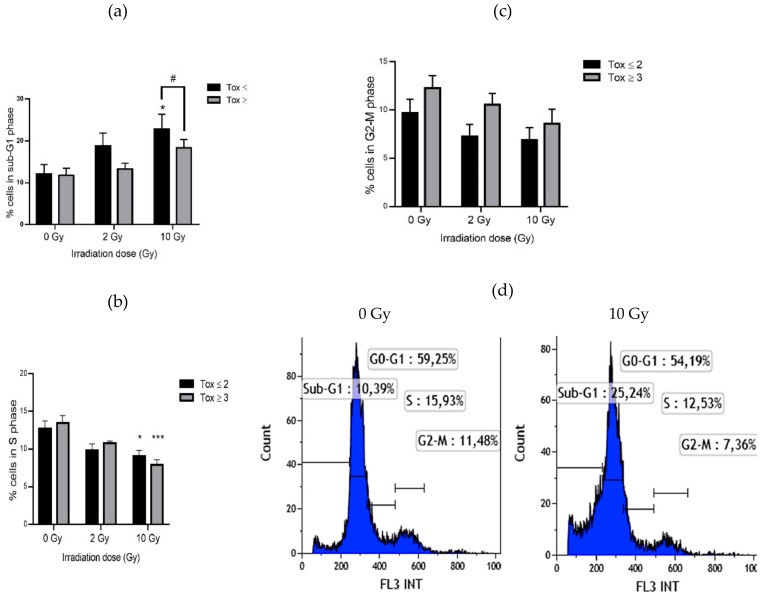
Lymphocyte percentage in sub-G1 phase (**a**), in S phase (**b**) and in G2-M phase (**c**) in both groups of patients and example of analysis of cell cycle for one patient (**d**). Groups Tox ≤ 2 and Tox ≥ 3 correspond to patients presenting grade 2 or less and grade 3 or more of late skin reactions after RT according to RTOG/EORTC, respectively. * for *p* < 0.05 and *** for *p* < 0.0001 for irradiated samples compared to non-irradiated ones and # for *p* < 0.05 for Tox ≥ 3 compared to Tox ≤ 2 group.

**Figure 3 antioxidants-09-00220-f003:**
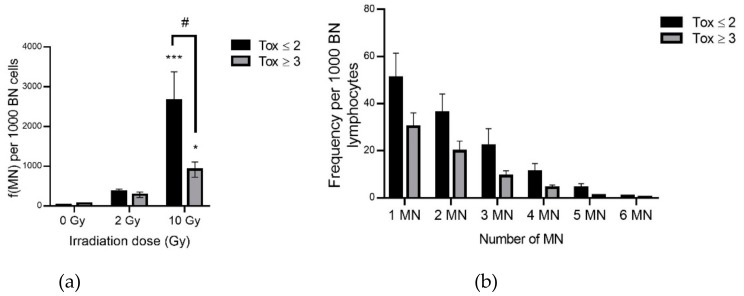
Micronucleus frequency (**a**) and distribution at 10 Gy (**b**) per 1000 binucleated lymphocytes in both groups of patients. Groups Tox ≤ 2 and Tox ≥ 3 correspond to patients presenting grade 2 or less and grade 3 or more of late skin reactions after RT according to RTOG/EORTC, respectively. * for *p* < 0.05 and *** for *p* < 0.0001 for irradiated samples compared to non-irradiated ones and # for *p* < 0.05 for Tox ≥ 3 compared to Tox ≤ 2 group.

**Figure 4 antioxidants-09-00220-f004:**
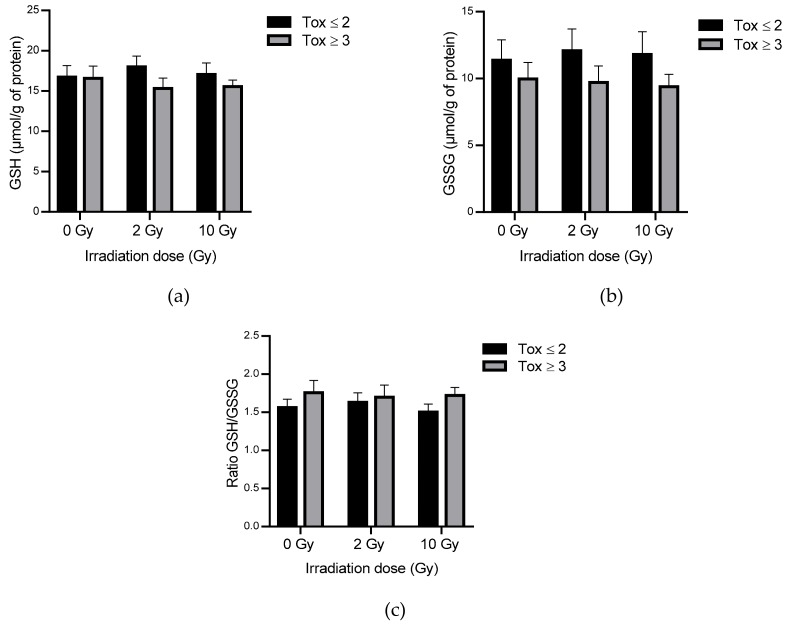
Reduced (**a**) and oxidized glutathione (**b**) and GSH (reduced glutathione)/GSSG (oxidized glutathione) ratio (**c**) in erythrocytes lysates in both groups of patients. Groups Tox ≤ 2 and Tox ≥ 3 correspond to patients presenting grade 2 or less and grade 3 or more of late skin reactions after radiotherapyaccording to RTOG/EORTC, respectively.

**Figure 5 antioxidants-09-00220-f005:**
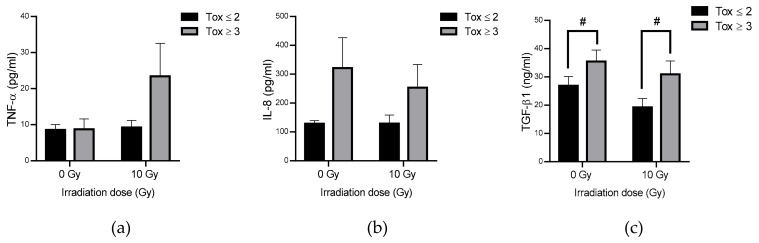
TNF-α (Tumor Necrosis Factor α) (**a**), IL-8 (Interleukin 8) (**b**) and TGF-β1 (Transforming Growth Factor β1) (**c**) concentration in plasma in both groups of patients. Groups Tox ≤ 2 and Tox ≥ 3 correspond to patients presenting grade 2 or less and grade 3 or more of late skin reactions after RT according to RTOG/EORTC, respectively. # for *p* < 0.05 for Tox ≥ 3 compared to Tox ≤ 2 group.

**Table 1 antioxidants-09-00220-t001:** Characteristics of the cohort depending on the toxicity group. Age, RTOG (Radiation Therapy Oncology Group)/EORTC (European Organization for Research and Treatment of Cancer) grade of late skin toxicity, total received dose, BED (Biological Effective dose) with α/β = 3, treatment year, gender and Fitzpatrick skin type are reported here. Groups Tox ≤ 2 and Tox ≥ 3 correspond to patients presenting grade 2 or less and grade 3 or more of late skin reactions after RT according to RTOG/EORTC, respectively.

Group	Patient Number	Age	RTOG	Dose (Gy)	BED (Gy3)	Treatment Year	Gender	Fitzpatrick Skin Type
Tox ≤ 2	1	80	0	40	93.33	2016	F	4
	2	75	2	50	83.33	2014	F	2
	3	61	1	50	83.33	2014	M	1
	4	76	1	44	102.67	2011	M	3
	5	77	1	50	83.33	2015	M	3
	6	74	1	50	83.33	2016	F	2
	7	56	0	50	83.33	2013	M	2
	8	91	1	48	86.40	2012	M	2
	9	71	0	50	83.33	2016	M	4
Mean ± SD		73.44 ± 3.62		48.00 ± 1.27	86.93 ± 2.39			
Tox ≥ 3	10	58	3	50	83.33	2015	F	2
	11	82	3	48	86.40	2006	M	2
	12	71	3	50	83.33	2016	M	2
	13	79	4	48	86.40	2010	F	3
	14	72	3	40	93.33	2014	F	3
	15	65	3	50	83.33	2011	M	2
	16	69	3	48	86.40	2013	F	2
	17	84	3	48	86.40	2015	M	3
	18	88	3	48	86.40	2015	F	2
Mean ± SD		74.22 ± 3.45		47.77 ± 1.09	86.14 ± 1.09			

**Table 2 antioxidants-09-00220-t002:** Micronucleus frequency per 1000 binucleated lymphocytes in both groups of patients. Groups Tox ≤ 2 and Tox ≥ 3 correspond to patients presenting grade 2 or less and grade 3 or more of late skin reactions after RT according to RTOG/EORTC, respectively. * for *p* < 0.05 and *** for *p* < 0.0001 for irradiated samples compared to non-irradiated ones and # for *p* < 0.05 for Tox ≥ 3 compared to Tox ≤ 2 group.

Irradiation Dose (Gy)	0		2		10	
Toxicity Group	Tox ≤ 2	Tox ≥ 3	Tox ≤ 2	Tox ≥ 3	Tox ≤ 2	Tox ≥ 3
f(MN) per 1000 BN cells	29.51 ± 8.16	61.68 ± 16.62	362.07 ± 65.81	280.56 ± 69.15	2663.13 ± 765.09 ***	916.31 ± 190.10 *#

**Table 3 antioxidants-09-00220-t003:** Micronucleus distribution at 10 Gy in both groups of patients. Groups Tox ≤ 2 and Tox ≥ 3 correspond to patients presenting grade 2 or less and grade 3 or more of late skin reactions after RT according to RTOG/EORTC, respectively.

Micronucleus Distribution at 10 Gy (Number of MN)		1 MN	2 MN	3 MN	4 MN	5 MN	6 MN
Toxicity group	Tox ≤ 2	47.22 ± 7.63	41.23 ± 9.55	26.46 ± 9.46	12.64 ± 4.59	5.35 ± 2.40	0.92 ± 0.38
	Tox ≥ 3	40.64 ± 8.49	22.95 ± 6.58	11.04 ± 4.33	4.82 ± 1.97	0.93 ± 0.36	0.16 ± 0.18

**Table 4 antioxidants-09-00220-t004:** Antioxidant enzyme activities in erythrocyte lysates in both groups of patients. Groups Tox ≤ 2 and Tox ≥ 3 correspond to patients presenting grade 2 or less and grade 3 or more of late skin reactions after RT according to RTOG/EORTC, respectively. Percentages of increase and decrease were calculated to compare Tox ≥ 3 vs. Tox ≤ 2 group. # for *p* < 0.05 and ## for *p* < 0.005 for Tox ≥ 3 compared to Tox ≤ 2 group.

Irradiation Dose (Gy)		0			2			10		
Toxicity Group		Tox ≤ 2	Tox ≥ 3	% Increase ↑ % Decrease ↓	Tox ≤ 2	Tox ≥ 3	% Increase ↑ % Decrease ↓	Tox ≤ 2	Tox ≥ 3	% Increase ↑ % Decrease ↓
Antioxidant enzyme activity	Superoxide dismutase (U/mg protein)	10.13 ± 1.2	9.23 ± 1.8	8.88% ↓	10.84 ± 1.45	7.36 ± 1.43	32.10% ↓ ^#^	10.01 ± 1.3	6.91 ± 1.2	30.97% ↓^##^
	Glutathione peroxidase (nmol of oxidized glutathione /min/mg protein)	41.59 ± 5.21	35.87 ±5.91	13.75% ↓	43.69 ± 5.41	38.47 ± 5.15	11.95% ↓	45.46 ± 5.35	38.28 ± 4.54	15.79% ↓
	Catalase (U/mg protein)	10.76 ± 1.90	8.89 ± 0.87	17.37% ↓	9.24 ± 0.51	8.89 ± 0.99	3.78% ↓	9.89 ± 0.84	9.11 ± 0.86	7.88% ↓

**Table 5 antioxidants-09-00220-t005:** Protein carbonylation and lipid peroxidation in erythrocyte lysates in both groups of patients. Groups Tox ≤ 2 and Tox ≥ 3 correspond to patients presenting grade 2 or less and grade 3 or more of late skin reactions after RT according to RTOG/EORTC, respectively. Percentages of increase and decrease were calculated to compare Tox ≥ 3 vs. Tox ≤ 2 group.

Irradiation Dose (Gy)	0			2			10		
Toxicity Group	Tox ≤ 2	Tox ≥ 3	% Increase↑ % Decrease↓	Tox ≤ 2	Tox ≥ 3	% Increase ↑ % Decrease↓	Tox ≤ 2	Tox ≥ 3	% Increase↑ % Decrease↓
Carbonyls (nmol carbonyl/mg of protein)	173.87 ± 21.5	180.86 ± 20.75	4.02%	175.22 ± 21.41	157.63 ± 15.55	10.04% ↓	176.73 ± 15.45	152.77 ± 13.8	13.56% ↓
Lipid peroxidation (nmol/ mg of protein)	21.79 ± 1.96	22.52 ± 1.52	3.35%	21.50 ± 1.81	23.45 ± 1.66	9.07%	21.45 ± 1.91	20.87 ± 1.02	2.70% ↓

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
