# Peer review of "The Possibility of Using Genotoxicity, Oxidative Stress and Inflammation Blood Biomarkers to Predict the Occurrence of Late Cutaneous Side Effects after Radiotherapy"

_antioxidants, 2020, doi:10.3390/antiox9030220_

Round 1

Reviewer 1 Report

Authors studied whether ex vivo blood sample could be a surrogate system to predict radiosensitivity of patients. Idea is excellent and the marker like micronucleus is interesting. But the small sample size seems problematic. To prove utility and validity of the markers author discovered in the screening of radiosensitive patients, proper prediction model needs to be established with the markers. And hopefully, with a different set of patients, the model shall be validated. (Furthermore, English writing needs significant polishing further and figures also need redrawing.

Reviewer 2 Report

In this work, the authors target the importance of radiotherpay (RT) associated toxicity which is mechanistically speaking related with the initial comlexity of damage as well as inflammatiory and immune response (higher) especially in most radiosensitive patients. The authors choose an ex-vivo approach where blood from a small size of patients is irradiated ex vivo and various endpoints are measured but suprisingly enough not DNA repair. They measure antioxidant capacity of erythrocytes (why not also lymphocytes?), micronuleus, inflammation panel but not DNA damage for example. The study is of high interest but there are some questions in the puzzle. My basic concern is that they use blood from RT patients and they do not know or describe at least with detail the background in all the parameters they measure. It is expected already these patients to have high levels of inflammation for example. Also the dose of 10 Gy is not logical. Why such a high acute dose? In addition, the DNA repair is not monitored where is expected to play a key role. 

Specific comments/concerns

  1. Late advances on the central role of complex DNA damage must be discussed since it is considered the key determinant of further biological response (please see Mavragani, I.V. et al. 2019) Ionizing Radiation and Complex DNA Damage: From Prediction to Detection Challenges and Biological Significance. Cancers, 11, 1789).
  2. The type of radiation and energies that were used is not described only the accelerator . Was in MV or KV range? More details are needed
  3. Again and based on comment 1, the systemic effects and this whole area of previous results are not discussed. What type of tumor was present in all cases. Different mechanistic studies using irradiation of a small part of the animal leads to an explosion of cytokines and inflammatory response. 

Author Response

First, we would like to thank this reviewer for helpful comments.

Among the panel of biomarkers studied on a small number of patients, DNA damage were quantified by micronucleus assay (lines 193-206 and 347-374, Figure 3 and Tables 2 and 3). Comet assay could not be performed for technical reasons as a large part of lymphocytes was for cell cycle analysis and the rest for micronucleus assay which seems to us more informative about DNA damage and repair, perhaps wrongly. As performed in other published papers about radiosensitivity prediction, there is a one hour incubation at 37°C after irradiation as described in Materials and Methods section: “At the end of the irradiation, samples were placed for 1 hour in a cell culture incubator at 37°C, 5% CO2 with a controlled humidity level” (lines 161-162). If we performed comet assay, it seems to us that we could miss DNA damage and repair after this incubation whereas micronuclei will be the reflect of DNA damage and mis- or un-repaired damage as mentioned in Results section: “Micronuclei contain a chromosome or fragment(s) of chromatid(s) which were not incorporated in one of the daughter nuclei after cell division. Micronucleus assay was used to evaluate incorrectly repaired or unrepaired DNA breaks or nondisjunction of chromosomes.” (lines 348-351).

Antioxidant capacity was measured on erythrocytes as they represent the main blood antioxidant capacity (with plasma antioxidants) knowing all plasma and lymphocyte samples were used for inflammation, cell cycle and genotoxicity. We added in the Discussion section: “Blood antioxidant capacity was assessed by the measurement of SOD, CAT, GPx, GSH and GSSG in erythrocytes as red blood cells are considered as the main actors of blood ROS elimination [52]” (lines 475-476).

Concerning background levels of each biomarkers, they are presented in each figures and tables as "0 Gy". As mentioned in the Discussion section, it is difficult to compare our results at basal level, we added more specifically: “Results obtained at the basal level (i.e. unirradiated blood) are difficult to compare with the data in the literature because of the different methods used and the variety of expression of the results (per volume, per mg of proteins or hemoglobin etc.). Although OS has been profusely studied in diverse pathologies as diabetes, Alzheimer's disease, and in aging, there is no validated data of rates of reference in blood.” (lines 480-484).

The dose of 10 Gy was chosen to eventually better discriminate patients by emphasizing the response after irradiation. The goal was not to be close to doses given during radiotherapy treatment.

Specific comments/concerns:

  1. We added the review suggested by the reviewer in discussion section with these sentences: "Moreover, as reviewed by Mavragani et al. [57], DNA damage/repair and chronic inflammation powered by the induction of DNA damage response are the most important factors to take into account for the prediction of radiation effects." (lines 517-519) and “Concerning DNA damage, Mavragani et al. [57] pointed out the major role of clustered DNA damage and the use of Monte-Carlo prediction model but also the interest of using bioinformatics and omics approaches.” (lines 541-543).
  2. The type of radiation and energie used was added in the Materials & Methods section: "Irradiation of blood samples was performed by 6 MV photon beams from an ARTISTE linear accelerator (Siemens) at room temperature in the RT department of Cancer Center François Baclesse" (lines 156-157).
  3. As appropriately suggested, we added in the Discussion section: "It is also known that localized irradiation can lead to a late increase in inflammatory status. As shown by Liao et al. on C57Bl6 mice 4 weeks after irradiation, IL1β and IL23 were strongly upregulated especially IL-17, which is responsible for upregulating gd T cells involved in mediating innate immune response, playing an important role on skin inflammation and thus on the appearance of radiation dermatitis [54].” (lines 503-508).

Reviewer 3 Report

Sir, 

I have reviewed the manuscript "Possibility of Using Genotoxicity, Oxidative Stress and Inflammation Blood Biomarkers to Predict the Occurrence of Late Cutaneous Side Effects after Radiotherapy" submitted by Samia Chaouni to Antioxidants with particular interest.  Radiotherapy remains a valuable therapeutic option in many diseases, including MCC. 

The Introduction is written well concerning radiobiology. However, the disease is poorly introduced. Merkel cešll carcinoma also has more therapeutic options that RT. Namely, Avelumab (trade name Bavencio) represents a recently widely used biological therapy, which is more efficient in advanced cases than RT. This must be acknowledged.

Further, the selection of MCC seems to be justified well because the RT is affecting the skin and dosimetry is precise. However, MCC arises on previously photodamaged skin in older people. This seems to be a rather unfavourable condition, because (photo-)ageing must also be taken into account as a critical factor for skin resistance to external stimuli (including RT). This aspect was entirely omitted by authors and I believe it must be actively commented in Introduction. 

Materials and Methods. 

  • relevant to  Patients: please, indicate whether dermatoporosis was assessed before RT initiation. This aspect can be clinically significant in response to RT.  Add this evaluation to Table  1 and also in section 3.1. Patient profiles 
  • relevant to Micronucleus frequency  - please, indicate magnification used in this study. Also, the Fig.3 would be more informative if the data table would accompany the graph. It is rather small for interpretation. 
  • relevant to 3.7. Inflammatory cytokines in plasma: just a comment, it is very difficult to believe that IL-6 was below the detection limit. This is one of the most abundant cytokines! 

Discussion

Authors conclude that their 18 patients showed a strong interindividual variability in radiosensitivity between patients. It is not surprising. Authors also correctly include suggestion regarding the skin biopsy. Skin ageing in such case seems highly relevant topic for further consideration (recently reviewed in Clinics in Dermatology by Strnadova and co-workers Clin Dermatol.

Skin aging: the dermal perspective. - PubMed - NCBI

pubmeddev

PubMed comprises more than 30 million citations for biomedical literature from MEDLINE, life science journals, a...

 2019 Jul - Aug;37(4):326-335 and also Zoubulis and co-workers in  Clin Dermatol. 2019 Jul - Aug;37(4):296-305. This should be actively commented. 

Also,  we should consider the increased levels of cytokines in serum as a consequence of ageing and notably also as a result of already existing malignant disease. It is also a warrant for any researcher that ageing in the context of radiotherapy is a critical feature. Authors absolutely correctly concluded: "Interestingly, the basal rate of plasmatic cytokines was
higher in radiosensitive patients suggesting that these patients present a higher inflammatory status than the least radiosensitive ones."  Yes, this is just indicating that these patients are more vulnerable due to ageing. However, this is a biological phenomenon and cannot be easily chronologically measured. This is the most critical conclusion of this study!! It is not a solitary observation, this is well-fitting into the complex change of paradigm of recent biological oncology. I guess that this aspect should be emphasised. 

I would suggest that professional language correction could improve legibility and increase attractiveness for readers. 

My overall impression of this manuscript remains positive and the text is worthy of attention after these (more or less minor) necessary corrections.

Round 2

Reviewer 1 Report

Authors made significant efforts to answer the reviewer's comments. Now it is acceptable for further process.

Reviewer 2 Report

The authors have improved considerably the manuscript and have shown a great respect to addressing all comments and concerns.